

# Dynamic Meteorology-Induced Emissions Coupler (MetEmis) development in the Community Multiscale Air Quality (CMAQ): CMAQ-MetEmis

Bok H. Baek[1], Carlie Coats[1], Siqi Ma[1,2], Chi-Tsan Wang[1], Jia Xing[1], Daniel Tong[1,2], Soontae Kim[4], and Jung-Hun Woo[3*]

[1]Center for Spatial Information Science and Systems, George Mason University, Fairfax, VA 22030, USA
[2] Department of Atmospheric, Oceanic and Earth Sciences, George Mason University, Fairfax, VA 22030, USA
[3] Civil and Environmental Engineering, College of Engineering, Konkuk University, Seoul, Republic of Korea
[4] Environmental Engineering, College of Engineering, Ajou University, Suwon, Republic of Korea

*Correspondence to*: Jung-Hun Woo (jwoo@konkuk.ac.kr)

## Abstract

The main focus of this study is to develop a dynamic-coupling "inline" air quality modeling system for the meteorology-induced emissions with simulated meteorological data. To improve the spatiotemporal representations and accuracy of onroad vehicle emissions, which are largely sensitive to local meteorology, we developed the "*inline*" coupler module called "MetEmis" for Meteorology-Induced Emission sources within the Community Multiscale Air Quality (CMAQ) version 5.3.2 modeling system. It can dynamically estimate meteorology-induced hourly gridded emissions within the CMAQ modeling system using modeled meteorology. The CMAQ air quality modeling system is applied over the continental U.S. for two months (January and July 2019) for two emissions scenarios: a) current "*offline*" based onroad vehicle emissions, and b) "*inline*" CMAQ-MetEmis onroad vehicle emissions. Overall, the "MetEmis" coupler allows us to dynamically simulate onroad vehicle emissions from the MOVES onroad emission model for CMAQ with a better spatio-temporal representation compared to the "*offline*" scenario based on static temporal profiles. With an instance interpolation calculation approach, the new "*inline*" approach significantly enhances the computational efficiency and accuracy of estimating mobile source emissions, compared to the existing "*offline*" approach that yields almost identical hourly emission estimation. The domain total of daily VOC emissions from the "inline" scenario shows the largest impacts from the local meteorology, which is approximately 10% lower than the ones from the "offline" scenario. Especially, the major difference of VOC estimates was shown over the California region. These local meteorology impacts on onroad vehicle emissions via CMAQ-MetEmis revealed an improvement in hourly $NO_2$, daily maximum ozone, and daily average $PM_{2.5}$ patterns with a higher agreement and correlation with daily ground observations.

Keywords: CMAQ, CTM, weather-aware emissions, vehicle emissions, inline modeling

## 1. Introduction

Since the industrial revolution, the chemical pollutants in the atmosphere have impacted human society due to their adverse health effects. The primary gases and particles directly emitted from their emission sources are chemically transformed into secondary pollutants through complex chemical reactions under various local meteorological conditions. Over last three decades, sophisticated multiscale chemical transport models (CTM) have been developed to predict the concentrations of primary and secondary chemicals in the lower





atmosphere, and actively used for air quality regulatory planning applications as well as for air quality forecasting for the general public health (Wong et al., 2012; Byun and Schere, 2006; Dennis et al., 2010; Rao et al., 2011; Hogrefe et al., 2001). The CTM simulation results strongly rely on two major inputs: meteorology and emissions, thus requiring accurate estimation of both to simulate the transport, chemical transformation, and removal of the pollutants. Depending on their chemical reactivity and gravitational

behaviors, some pollutants can be chemically transformed and travel a long distance from their source of origin while some are deposited near their release locations.

To accurately predict regional and global chemicals in the future, spatially and temporally resolved meteorology and emissions are critical and required to be rapidly updated based on the aerosol direct/indirect meteorology impacts within a fully coupled air quality modeling system. There have been considerable

amounts of efforts in meteorology prediction enhancements actively conducted (Jacob and Winner, 2009; Grell and Baklanov, 2011; Fiore et al., 2012; Wong et al., 2012). However, there have been only limited "*inline*" emissions modeling enhancements made to CTM system wherein emissions from meteorologically driven air pollutant emission processes are dynamically coupled within the regional/global CTM modeling system, rather than being estimated *a priori* and statically provided as model inputs based on "offline" spatial

and temporal allocations. Simulating emissions "*inline*" is especially crucial for real-time air quality forecasting (Tong et al., 2012). In particular, the system of the National Oceanic and Atmospheric Administration (NOAA) National Air Quality Forecast Capability (NAQFC) allows to induce the influences of the forecast meteorology on emissions from key sources, such as stationary power plants, vegetation, fertilizer applications, such as mineral dust (Knippertz and Todd, 2012), sea salt (Foltescu et al., 2005; Pierce

and Adams, 2006), biogenic volatile organic compounds (BVOCs) (Lathière et al., 2005; Chen et al., 2018), and biomass burning events (Grell et al., 2011; Pavlovic et al., 2016). Despite these scientific advancements and model improvements, true process-based interaction between local meteorology and meteorology-induced anthropogenic pollutant emissions from onroad vehicles, livestock wastes, and residential heating remain incomplete or overlooked (Pouliot, 2005; Tong et al., 2012).

The mobile/transportation sector is one of the most important anthropogenic emissions sectors in metropolitan regions where most of high ozone and $PM_{2.5}$ concentration episodes often occur (Andrade et al., 2017; Kumar et al., 2018; Perugu, 2019). It is also known that the performance and emissions of mobile engines are sensitive to local weather conditions, such as ambient temperature and humidity (Lindhjem et al., 2004; Iodice and Senatore, 2014; Choi et al., 2017; Mellios. et al., 2019). The incomplete fuel combustion

can be occurred under cold ambient temperature and high humidity, leading to higher emissions emitted. The effect of humidity on internal combustion engines, including spark-ignition engines (gasoline, LPG, and natural gas) and compression ignition or diesel engines, has been known for many years, with evidence indicating that higher humidity results in lower NOx emissions(Lindhjem et al., 2004; USEPA, 2015). Additional emissions also come from energy usage of air conditioning at higher ambient temperatures. These

meteorological impacts can be accounted for using the state-of-science mobile emissions models such as the U.S. EPA's MOtor Vehicle Emission Simulator (MOVES) version 3.0 (USEPA, 2020). However, it lacks transparency of air pollutant emission algorithms, including key parameters such as emission factors. Furthermore, it requires significant computational resources to generate these high-quality spatiotemporal emissions from onroad vehicles (Li et al., 2016; Xu et al., 2016; Liu et al., 2019; Perugu, 2019). To provide

the weather-aware onroad mobile emissions to the current CMAQ, the MOVES has been integrated with the Sparse Matrix Operator Kernel Emissions (SMOKE) modeling system (Baek et al., 2010) by processing (reading/storing/accessing) MOVES emission factors (EF) datasets. However, it demands a significant computational time and memory due to the high traffic of input/output (I/O) data, which largely prohibits its





usage in real-time air quality forecasting. As an example, the latest version of SMOKE version 4.8.1 can

require >3 computing hours with up to 20GB RAM memory to generate 25 hours CMAQ-ready gridded hourly emissions over Continental U.S. (CONUS) modeling domain (12km *12km grid size).

To enable the direct feedback effects of aerosols and local meteorology in an air quality modeling system without any computational bottleneck, we have developed an "*inline*" meteorology-induce emissions coupler module within the US EPA's CMAQ modeling system, called "Meteorologically-induced anthropogenic

Emissions: CMAQ-MetEmis", to dynamically model the complex MOVES onroad mobile emissions inline without a separate dedicated emissions processing model like SMOKE. To address the shortcomings (computational time and memory requirements) in the current slow "*offline*" approach, we first re-restructured the current CMAQ-ready surface gridded hourly emissions output from SMOKE into the ambient temperature-specific gridded hourly emissions and store them into a pseudo-layer structure for easy

and fast access. Each pseudo-layer holds the gridded chemically-speciated hourly emissions by incremental temperature bin (e.g., 10F, 20F, and so on). The CMAQ-MetEmis coupler was developed to estimate the gridded hourly emissions with a simple linear interpolation between two temperature-bins gridded hourly emissions based on a simulated hourly ambient temperature. With an instance interpolation calculation approach, the new "*inline*" approach significantly enhances the computational efficiency compared to the

existing "*offline*" approach without losing any accuracy of emission estimates. We also evaluate the performance of the CMAQ-MetEmis coupler module in CMAQ which includes their computational performance, the feasibility of CMAQ-MetEmis implementation as a forecasting application, the responses of $O_3$ and $PM_{2.5}$ to the meteorological impacts on anthropogenic emissions.

## 2. CMAQ-MetEmis Development

NOAA has developed the NAQFC, operated by the National Weather Service (NWS), in partnership with the U.S EPA using the state-of-science air quality modeling system, CMAQ, to forecast concentrations of $O_3$ and $PM_{2.5}$ over the contiguous continental U.S. (CONUS), Alaska and Hawaii (Tong et al., 2015; Lee et al., 2017; Tang et al., 2017). Unlike weather forecasting, air quality forecasting requires full atmospheric chemistry along with the physical state and tendency of the weather in the near future. Accurate prediction

of meteorology and emissions for CMAQ plays a critical role in the accuracy of 48- and 72-hour air quality forecasting. The current NOAA/NWS operational requirements specify that the post-processing of the simulated/forecasted meteorological data, emission data, and air quality chemistry model simulations be completed in a reasonable time frame to meet the air quality forecasting time constraints. Since the processing of the meteorological data and the execution of the air quality chemistry model are the most time-

consuming part of CMAQ, minimizing the processing time of the emissions needs is desirable. A typical emission-processing over U.S. CONUS national domain for one day may take up to 2 hours on a single CPU (Intel Xeon Gold 6240R @ 2.4GHz) using SMOKE and other post-processing tools. To expedite the operational forecasting streamlines, non-meteorological dependent emissions are generally processed in advance (Tong et al., 2015). Only the meteorologically induced emission sources are processed during the

air quality forecasting simulation runs. So then, the accuracy of the emission processing can be maintained, and the forecast can be completed within the required time constraints. However, due to the high computing CPU hour requirement to estimate the high-quality onroad mobile emissions from MOVES, the SMOKE-MOVES integration tool that allows dynamically estimating weather-aware gridded hourly emissions with the forecast meteorology from NAQFC has not been implemented in the current NAQFC operations (Tong

et al., 2015).



## 2.1 Modeling Configuration

The George Mason University (GMU) air quality modeling system in this study is configured close to the current operational NAQFC, including the spatial coverage, emission inputs, and chemical transport model.
It contains three major components: meteorology, emission, and chemical transport models.. The Weather Research and Forecasting (WRF) model version 4.0 is used to generate hourly meteorological fields to drive emission and air quality modeling. The WRF model was configured with Thompson graupel microphysics scheme, RRTMG long and short-wave radiation scheme, Mellor-Yamada-Janjic PBL scheme, unified Noah land-surface model, and Tiedtke cumulus parameterization option. The emission input was provided using a
hybrid emission modeling system that utilized the SMOKE model version 4.8.1 (Baek and Seppanen, 2021) to process anthropogenic emissions, and a suite of emission models to estimate emissions from intermittent and/or meteorology-dependent sources. Anthropogenic emissions were taken from US EPA 2017 NEI. The CMAQ model (version 5.3.2) ingests emissions and meteorology to predict spatial and temporal variations of the atmospheric pollutants (such as $O_3$, $NO_2$, and particulate matters) using a revised Carbon Bond 6 gas-
phase mechanism and AE7 aerosol mechanism (CB6r3_AE7_AQ) (Byun and Schere, 2006; Luecken et al., 2019).

The meteorological, emission and air quality models have 12×12 km horizontal resolution over the contiguous United States, with full 35 sigma layers vertically and the domain top at 50 hPa. The WRF model was driven by the forecast fields of Global Forecast System (GFS) version 4 products with a horizontal
resolution of 0.25° × 0.25° (available every 6 h) and was reinitialized every 24 hr to be consistent with its operational task.

To understand the impacts of meteorology-induced onroad emissions on local air quality, we conducted two CMAQ simulation scenarios. All simulations were conducted for two months, January and July in the year 2019. We initiated our CMAQ simulations based on the default CMAQ background concentration profiles.
The first three days of CMAQ simulation were used as a spin-up modeling period to eliminate the influence of the initial condition (Chen et al., 2021; Lv et al., 2018; Tong and Mauzerall, 2006). The configurations and simulations are listed in Table 1.

1. "Base" scenario: Static gridded hourly emissions based on the county total emissions with static
155         temporal profiles (monthly, weekly, month-to-day, and hourly).
2. "*MetEmis*" scenario: Weather-aware gridded hourly emissions dynamically simulated with simulated meteorology using the inline "CMAQ-MetEmis" approach.

## 2.2 Meteorology-Depended Mobile Emissions

Mobile emissions from onroad and off-network (e.g., vehicle start-up, running exhaust, break-tire wear, hot soak, and extended idling) are much sensitive to temperature and humidity due to various factors, 1) cold engine starts that enhance emissions at lower ambient temperatures due to the incomplete fuel combustion, 2) evaporative losses of volatile organic compounds (VOCs) due to expansion and contraction caused by ambient diurnal temperature variations, 3) enhanced running emissions at higher ambient temperatures, 4)
atmospheric moisture suppression of high combustion temperatures that lower nitrogen oxide emissions at higher humidity, and 5) indirect increased emissions from air conditioning at higher ambient temperatures (Choi et al., 2017; Iodice and Senatore, 2014; Lindhjem et al., 2004; Mellios. et al., 2019; USEPA, 2015). McDonald *et al*. (2018) found that NOx emissions from NEI estimated from the U.S. EPA's MOVES are





under-estimated, leading to a failure of prediction of high ozone days (8-hr max ozone>70 ppb). (McDonald
et al., 2018)

The dependency of mobile emissions on local meteorology can vary by vehicle types (light-duty, heavy-duty, truck and bus), fuel types (gasoline, diesel, hybrid, and electric), road types (interstate, freeway, local roads), processes (vehicle start-up, running exhaust, break-tire wear, hot soak, and extended idling), vehicle speed for onroad vehicles, hour of day for off-network vehicles, as well as by pollutants such as CO, $NO_X$, $SO_2$,
$NH_3$, VOC, Particulate Matter (PM). Figure 1 shows the dependency of MOVES emission factors of CO, NOx, VOC, and $PM_{2.5}$ from gasoline-fueled vehicles on ambient temperature from onroad and off-network, respectively. All pollutant emissions vary with the temperature, particularly under lower speed. The CO, VOC and NOx emissions increase with the temperature while opposite relationship is suggested between $PM_{2.5}$ emissions and temperature, implying the complexity of meteorology impacts on different pollutant
emissions. For off-network emissions from gasoline-fueled vehicles, CO, NOx and $PM_{2.5}$ show negative correlations with temperature, while the VOC exhibits nonlinear response to the temperature variation. The largest meteorology dependency occurs in daytime when emissions are the greatest across a day. Further detailed meteorology dependency of MOVES emission factors on local meteorology can be found in Choi *et al.*, 2017.

**2.3 SMOKE-MOVES Integration Tool**

In 2010, U.S. EPA introduced the process-based onroad mobile emissions model, MOVES, which is a state-of-the-science MySQL database-driven software for calculating bottom-up vehicular emissions from onroad and off-network. Off-network emission processes (e.g., parked engine-off, engine starts, and idling, and fuel vapor venting) in MOVES are hour-dependent due to vehicle activity assumptions built into the MOVES
model; the emission rate in a unit of grams/mile/hour depends on both hour of the day and temperature. Onroad emission processes (e.g., running exhaust, crankcase running exhaust, brake wear, tire wear, and on-road evaporative), on the other hand, do not depend on the hour but are expressed in grams/mile.

MOVES is approved for use in official state implementation plan (SIP) submissions to EPA and for conformity emissions inventory development outside of California. Furthermore, it can be used to estimate
onroad vehicle emissions for a variety of different purposes: to evaluate the national and local emissions trends, to compare different emission scenarios, to analyze the benefits from mobile source control strategies, and to provide inputs for air quality modeling. Although MOVES estimates of mobile emissions include the dependence on vehicle activities and simulated hourly meteorology, its computational requirements are prohibitive in real-time air quality forecasting applications. To overcome these issues, the SMOKE-MOVES
tool was developed by integrating MOVES emission factor (EF) outputs with the SMOKE modeling system (Baek, 2010), with the objectives of reducing processing time, and improving the accuracy of mobile emissions for air quality modeling. The tool allows hourly mobile emissions estimates based on vehicle activity inventories (i.e., miles traveled, population, and operating hours), MOVES EFs (a function of vehicle type, road type, and local meteorology), and simulated hourly ambient temperatures, and humidity. It first
estimates spatially and temporally averaged county-level hourly meteorological inputs (temperatures and humidity). It then prepares driver and post-processing scripts to set up and run MOVES to generate county-specific MOVES EF lookup tables (LUT), and to sort them by average vehicle speed, ambient temperatures, humidity, operating hours, day of week, and/or hour of the day. Finally, the tool runs SMOKE to estimate air quality model-ready emissions using the MOVES EF LUTs with hourly meteorological inputs.
Based on the latest 2017 National Emissions Inventory (NEI) Emissions Modeling Platform (EMP) (USEPA, 2022), the county-specific individual MOVES EF LUT file size can range from 60MB up to 150MB, and



processing so many MOVES EF LUT files from the targeted counties in our modeling domain (e.g., 12kmx12km grid over U.S. Continental) require significant computational resources, such as memory and storage spaces.

Development of CMAQ-MetEmis Coupler One of the future key advances for current CMAQ in NAQFC application is developing a unified forecast system (UFS) with dynamically coupled process-based emissions modeling to provide atmospheric chemicals feedbacks to climate and meteorology, and to boost the air quality forecast modeling applications in seasonal-to-sub seasonal predictions. While biogenic emissions, bi-directional $NH_3$ from fertilizer applications, and point-source plume rise are dynamically coupled in CMAQ

"*inline*" as a part of NAQFC, these known meteorology-induced emissions sectors have little or no accounting of meteorological impacts in current operational chemical and aerosol forecasts but are represented with static, no-weather-aware annual or monthly county total emissions and standard monthly/weekly/daily temporal allocation profiles to disaggregate them on finer time scales for the hourly air quality forecasts. It often results in poor forecasting performance due to the poor spatiotemporal

representations of precursor pollutants during high ozone and $PM_{2.5}$ episodes (Tong et al., 2012).

In this study, we developed the meteorologically-induced emissions coupler module (MetEmis) within the CMAQ modeling system to enhance the current NAQFC with the weather-aware emissions modeling capability without any additional computational burden to the system. Pouliot (2005) indicated that the main obstacle to implementing weather-aware emissions into air quality simulation is a significant computational

resource requirement, especially for air quality forecasting applications. To address these potential shortcomings (computational time and memory requirements), we first implemented a new feature in the SMOKE v4.8.1 modeling system to generate the temperature-specific pre-gridded hourly emissions called "MetEmis_TBL", and then store them into the pseudo-layer structure for easy and fast access for later weather-aware emissions estimations (Figure 2). Each pseudo-layer holds the pre-gridded hourly emissions

based on pre-defined temperature bins (e.g., 5°C, 10°C, 15°C, and so on).

There are two ways to process the "MetEmis_TBL" emissions input file to develop weather-aware emissions: (a) "SMOKE-MetEmis", and (b) "CMAQ-MetEmis". The "SMOKE-MetEmis" is an "*offline*" approach based on the updated SMOKE modeling system with the "MetEmis_TBL" that can dynamically estimate weather-aware gridded hourly emissions with the forecast meteorology prior to the CMAQ simulations

(Figure 2a). The updated *Mrggrid* utility tool from the SMOKE will first read and process the "MetEmis_TBL" emissions file with the forecast meteorology as a part of the emissions processing step prior to the CMAQ simulations. However, the "CMAQ-MetEmis" is a true "*inline*" approach based on the CMAQ version 5.3.1 with a new dynamic emission coupler module called "MetEmis" that can internally generate weather-aware emissions with "MetEmis_TBL" within the CMAQ simulations (Figure 2b). While both

approaches generate the same CMAQ ready gridded weather-aware hourly emissions, the "CMAQ-MetEmis" approach will not only require any offline emissions modeling using SMOKE, but also expedite its computational processing time with the CMAQ parallelized simulations.

## 3. Results

In order to evaluate the impact of the "*MetEmis*" approach, the CMAQ modeling system is performed for

two different scenarios, "*MetEmis*" and "*Base*", respectively for the winter (January) and winter (July) seasons of 2019. First, we analyze the response of NOx, VOC, $NH_3$, and $PM_{2.5}$ emissions to the dynamic "inline" SMOKE-MetEmis approach. Then, the evaluation of the CMAQ-MetEmis air quality modeling system is performed by the comparison of the simulated ambient concentrations of $NO_2$, $O_3$, and $PM_{2.5}$ with





the observations where the most of meteorology-induced emissions are impacted by the meteorology
compared to the "offline" static approach (i.e, Base).

### 3.1 Computational Efficiency

While estimating meteorologically-induced onroad mobile emissions using local meteorology accurately
provides the emissions to CTM, the current "*offline*" SMOKE-MOVES integration tool approach has faced
many challenges, such as computational burdens, and the data portability and distributions due to the size of
data files and computationally expensive I/O data processing. Accurately generating the onroad mobile
emissions for the U.S. continental using MOVES onroad emission model requires a significant amount of
computational resources as well as the processing time. It approximately takes 12 computing hours to
generate one county MOVES EF LUT table per month using MOVES (Baek et al., 2010). Simulating over
3,100 counties in the U.S. continental (CONUS) for 12 calendar months (>37,400 MOVES simulations) will
require a tremendous amount of computational resources and time. Thus, U.S. EPA has adopted the
representative county approach to reduce the number of counties as well as the number of modeling months.
Each representative county was classified according to its state, altitude (high or low), fuel region, the
presence of inspection and maintenance programs, the mean light-duty age, and the fraction of ramps (CRC,
2019). A total of 296 representative counties for CONUS and 38 for Alaska, Hawaii, Puerto Rico, and the
US Virgin Islands (USEPA, 2022). Each representative county holds two fuel months to represent all 12
calendar months. Based on the 2017 NEI EMP, the county-specific individual MOVES EF LUT file size can
range from 60MB up to 150MB, and there are a total of 668 MOVES EF LUT input files which represent
3,100 counties in the U.S. for an entire modeling year (334 LUT files per fuel month).
To generate one day (25 hourly time steps) CMAQ-ready gridded hourly emissions, SMOKE needs to read
and process 334 MOVES EF LUT as well as many other SMOKE-ancillary input files such as VMT activity,
temporal profiles, chemical speciation profiles, spatial surrogates, and so on. The most computational
resources are consumed in I/O (inputs and outputs) of huge amount of data files while it processes the
complex datasets. Table 2 shows the estimated computational resources and time per each onrad mobile
sector (e.g., RatePerDistance (RPV), RatePerVehicle (RPV), and RatePerHour (RPH)). Among the mobile
sectors, RPD and RPV are the slowest sectors processed in the SMOKE modeling system.
Based on the latest 2017 NEI EMP, CMAQ-ready gridded daily emissions in our modeling domain (e.g.,
12kmx12km grid over U.S. Continental) requires approximately 1.9 hours per day (RPD: 90 minutes, RPV:
18 minutes, and RPH: 1 minute) to generate the complete set of onroad mobile daily emissions including
RPD, RPV and RPH modes. It may require over 638.5 hours (~29 days) of computational time to generate
CONUS gridded hourly emissions for 365 days.  While the CMAQ-MetEmis "*inline*" approach (Figure 2b)
does not cause much computational processing time since the I/O of NetCDF/IOAPI binary format
MetEmis_TBL input file in the CMAQ modeling system is instantaneous. There was less than 1 minute per
day of CMAQ computational time with 96 CPUs parallel processing.
The SMOKE-MetEmis can generate a single MetEmis_TBL emissions input file that holds the 25
temperature-bins gridded hourly emissions for 334 representative counties for one fuel month from 0°F to
125°F temperature (25 bins with 5°F increment). Correction equations for humidity are applied to estimate
grid-cell-hour adjustment factors for $NO_x$ emissions by fuel type (USEPA, 1997). The size of MetEmis_TBL
input file that can represent the 334 MOVES LUTs files per fuel month with 25 temperature bins is
approximately 16GB which is significantly smaller than the size for 334 MOVES LUTs files, ~ 62.8GB.





Approximately 6 hours are required to generate the MetEmis_TBL file once with SMOKE per fuel month, prior to the CMAQ-MetEmis simulations.

### 3.2 Weather-Aware Mobile Emissions

The huge computational burden of traditional "offline" SMOKE-MOVES approach prohibits its usage in providing real-time estimates of mobile emissions which might be significantly driven by the weather changes, resulted in considerable uncertainties in predicting emissions and air quality. The spatial monthly total difference plots of VOC and NOx between "Base" and "MetEmis" from Figure 3 clearly show that most of the emission differences caused by local meteorology occur from major interstate roads and metropolitan

cities (e.g., New York, Detroit, Chicago, Los Angeles, Phoenix, and Atlanta), where onroad mobile emissions contribute the most. Especially, the most differences in VOC were occurred over California region in July 2019, probably because the original temporal profiles assumed in "Base" are not suitable to represent the real condition influenced by the weather. The January and July VOC emissions from the "Base" scenario were higher by over 8% and 20% than the ones from the "MetEmis" scenarios, respectively, indicating that current

NAQFC-ready onroad mobile emissions (no-weather-aware) are significantly over-representing the VOC emissions compared to the weather-aware VOC dynamically estimated by MetEmis.

Unlike the "Base" approach, the "MetEmis" approach estimates hourly emissions by multiplying the estimated hourly vehicle mileage traveled (VMT) in the unit of miles/hour with inventory pollutant emission rates (unit of grams/miles), which are a function of local meteorology (e.g., ambient temperature and

humidity). The "MetEmis" emissions can enhance their spatiotemporal representations of onroad mobile sources. However, the hourly VMT activity data is estimated using the same temporal profiles used in the "Base" hourly emissions. Thus, both onroad emissions follow similar weekly and daily patterns with some hourly variations based on local meteorological conditions. As presented in Figure 4 which compares the hourly domain total TOG (Total Organic Gases), NOx, and $PM_{2.5}$ emissions between the "Base" and the

"MetEmis" approach, the statically estimated "Base" hourly emissions (colored blue) clearly show the repeated weekly patterns within the same month due to the usage of the static weekly temporal profiles, while the "MetEmis" (colored in red) display irregular hourly patterns due to the impacts of local hourly meteorology.

Due to the influence of local meteorology (i.e., ambient temperature and relative humidity), the onroad

running exhaust/evaporative emissions from RPD and the off-network evaporative emissions from RPV modes shows a moderate decrease of TOG and a slight increase of NOx (> 4% increase) over the entire domain due to low ambient and humidity condition during the winter season (January), according to "MetEmis" estimates. The most important enhancement in "MetEmis" approach is that it allows modelers to simulate NAQFC-ready weather-aware onroad mobile emissions. More important, the daily differences are

also noticeable in "MetEmis" approach within one month, as higher TOG and $PM_{2.5}$ are shown in late January due to the increased temperature, while the "Base" approach failed to predict such variation. Such spatiotemporal enhancements of onroad mobile emissions predicted by "MetEmis", especially near metropolitan regions, would benefit the NAQFC.

### 335   3.3 Effects of Weather-Aware Mobile Emissions on simulations

This study investigated the response of $NO_2$, $O_3$ and $PM_{2.5}$ to the meteorology-induced mobile emission changes by simulating air quality under two scenarios (Base and MetEmis). The sensitivity of air pollutant concentrations to these meteorology-induced emission sources was performed and analyzed in this section.





The monthly statistical modeling evaluation metrics for these two simulations (Base and MetEmis) over the CONUS domain are provided in Table 3. The correlation coefficient (CORR) of $O_3$ is 0.51 for both simulations, and they have the same normalized mean bias and errors (NMB and NME), while the relative mean square error (RMSE) of Base (7.03 ppb) is slightly higher than that of MetEmis (7 ppb). The simulated $NO_2$ shows the best correlations (0.64) among these three pollutants in January, however, its RMSE, NMB, and NME are the largest. The $PM_{2.5}$ simulation didn't reproduce the variability very well with a lower CORR of 0.46, but it presents the best RMSE and moderate NMB/NME. In July, the CORRs of $O_3$ improve from 0.51 to 0.64, while the RMSEs are also increasing because of intense concentration in summer. The $NO_2$ and $PM_{2.5}$ have the opposite pattern compared to that of $O_3$ with decreased CORR (0.51 and 0.38, respectively) and improved biases and errors, except the NME of $NO_2$. Over the entire modeling domain, both simulations show quite similar modeling performances against the observations, with the difference generally below 1%. This is mostly attributable to the spatial pattern of emissions which is primarily concentrated in urban areas. The most impacts of MetEmis emissions are shown over metropolitan cities where mobile emissions play a critical role in their local air quality.

Figure 5 shows the monthly average $NO_2$, $O_3$, and $PM_{2.5}$ concentrations from the Base scenario and the monthly average difference between the Base and MetEmis scenarios in July 2019. The spatial distributions of simulated $NO_2$ present a close pattern with those of $NO_x$ emission in both two months, demonstrating the effect of local $NO_x$ emission on the $NO_2$ activities. The $NO_2$ concentration in July is lower than January, which is caused by the stronger $NO_2$ photolysis and ventilation. In January, the $NO_2$ simulated by MetEmis showed higher concentration over the domain with more than 0.2 ppb larger over urban areas because of the increased $NO_x$ emission after adjustment. In comparison, the monthly simulated $NO_2$ concentrations with and without emission adjustment are much closer in July, the emission adjustment makes the concentration increase in the east while a decrease in the west. Compared to $NO_2$, the secondary $O_3$ and $PM_{2.5}$ formation chemical reactions involve complex nonlinear processes under various meteorological conditions and precursor emissions. Despite their complexity, there are strong correlations between their nonlinear responses and precursor emission changes.

The $O_3$ concentration is generally below 36 ppb in most areas in January because of the cold weather and weak photolysis process, while it presents high over the mid-western US which is caused by the higher altitude over the Rocky Mountains area. The $O_3$ significantly increases in July with average concentration of 43.9 ppb, which is 10 ppb larger than that in January. In July, the northeastern US becomes the hot spot zone as the local anthropogenic emission and pollution transport are both strong. In the meanwhile, the $O_3$ also concentrated over the water, such as Great Lake and northeastern coastal areas. The most of ozone increase occurred around the surrounding regions of metropolitan cities like Chicago, IL, Atlanta, GA, Denver, CO and Pheonix, AZ, where both NOx and VOC emissions are slightly increased during July 2019 (Figure 3). However, San Jose area shows a significant decrease of ozone during the summer in 2019 due to the higher VOC estimations from NEI (Base) compared to the ones from MetEmis scenario (Figure 3).

The $PM_{2.5}$ simulation has similar patterns in January and July with more particles concentrating in the east. The southwestern areas show less particulate pollution as the emission we use does not include natural sources such as dust storms and wildfires. The results from MetEmis present slightly higher $PM_{2.5}$ in the east because of the increased primary $PM_{2.5}$ emission. In addition, a decreased $PM_{2.5}$ concentration is noted in California. This may attribute to the less generated secondary aerosols as the VOC emission is significantly reduced after adjustment.



### 3.4 Evaluation on Modeling Performance

This study further examines the influence of meteorology-induced mobile emission changes on modeling performance which is particularly important for the air quality forecasting in NAQFC. 10 cities with the most changes in emissions are selected for comparison, as shown in Figure 6. In general, noticeable

improvement is found in $NO_2$ simulation with increase $R^2$ in all 10 cities except Detroit. San Jose and Atlanta exhibits the largest improvement in $NO_2$ simulation. Apparently, the MetEmis successfully captured daily variations of mobile emissions, resulted in an improved temporal correlation. Meanwhile, the RMSEs were reduced in most of cities (8 out of 10), suggesting the simulated biases can also be eliminated with MetEmis.

Compared to $NO_2$, changes in $O_3$ and $PM_{2.5}$ are smaller due to the complex reactions. However, improvement is also found in summer with increased $R^2$ and reduced RMSE in more than 70% cities, though less improvement is suggested in winter. We analyzed a few episodes with the largest changes for $O_3$ and $PM_{2.5}$ to better demonstrate such improvement.

**Ozone Episodes Analysis**

Based on the July 2019 CMAQ simulation between the Base and MetEmis cases, we identified the locations where the largest changes in surface ozone occurred. Especially, in July 2019, we witnessed a significant decrease in ozone over San Jose, CA at 1:00 PM local time on July 24, 2019, while the most increase in ozone occurred over Chicago, IL at 11:00 AM on July 5, 2019 (Table 4). Thus, we investigated

these two episodes to understand what the main drivers of these behaviors are.

*Largest Ozone Increase Episode*

Figure 7 shows the spatial ozone concentrations and the differences over Chicago region between the Base and MetEmis scenarios at 11AM LST on July 5, 2019. While the highest ozone occurred around the south

of Michigan lake in both scenarios (Figure 7a), the largest ozone increase (~7ppb) is shown in the middle of Michigan lake, where unfortunately there is no AQS monitoring location (Figure 7b). To understand the cause of these ozone changes, we examined the differences of NOx and VOC emissions between Base and MetEmis scenarios. The increase of VOC emissions from the MetEmis scenario in the early morning (3LST-9LST) over the VOC limited Chicago, IL region seems to be the main driver of a significant

increase of ozone (Figure 8). The detailed information on VOC and NOx concentration changes on July 5[th], 2019, is listed in Table 5. In the early morning, there was a decrease in NOx concentration, and an increase in VOC concentrations over Chicago area. Due to no monitoring location available over the lake, we were not able to properly perform the modeling evaluation statistics during the largest ozone increase.

*Largest Ozone Decrease Episode*

There was more than an 80ppb ozone decrease over San Jose, CA at 11LST on July 24[th], 2019. To understand the cause of this significant decrease, we performed the analysis of precursor emissions changes during the episode period. The colored green AQS locations are selected for the ozone concentration analysis, while the red ones are for the $PM_{2.5}$ monitoring locations (Figure 9a). Figure 9b shows the

modeled hourly ozone concentrations (maximum, minimum, and mean) and AQS observations over the blue box targeted region from Figure 9a. Figure 9b and Figure 10 indicate that the maximum ozone values from "Base" scenario clearly show an overestimated ozone over San Jose, CA downwind region, while the MetEmis case shows a significant improvement in maximum ozone concentration during the daytime. The



main driver of this significant ozone change over the San Jose targeted area is due to the substantial
reduction in VOC emissions in MetEmis from Base (Figure 11a). Statistics of NOx and VOC
concentrations from CMAQ in Table 6 show consistent findings.

### *Largest PM$_{2.5}$ Decrease Episodes*

Along with the significant ozone decrease in July, 2019, there was a significant PM$_{2.5}$ decrease from
CMAQ-MetEmis simulation from 42.5μg/m$^3$ (Base) to 25μg/m$^3$ at 10LST on January 3, 2019.
Approximately 17.5 μg/m$^3$ (>41%) PM$_{2.5}$ decrease was witnessed in CMAQ-MetEmis simulations (Figure
11). The CMAQ-MetEmis simulation shows a significant improvement in modeled PM$_{2.5}$ concentration,
compared to the one from the AQS monitoring locations from 8a (Figure 12a). The main cause of this
PM$_{2.5}$ decrease in CMAQ-MetEmis is mainly a significant decrease in primary PM$_{2.5}$ and VOC emissions
(Figure 13). Primary hourly PM$_{2.5}$ emissions from MetEmis scenario were significantly lowered than the
ones from Base scenario, approximately a maximum of 20kg/hour from 3LST to 9LST on January 3, 2019.

## 4. Conclusions

To address the limitation of traditional estimation for onroad vehicle emissions, this study developed a novel
method (*i.e.*, MetEmis) by dynamically coupling the meteorology-induced onroad emissions with simulated
meteorological data in the air quality modeling system, which significantly improves both computational
efficiency and accuracy. The computational time for processing one day onroad emission data is substantially
reduced from 1.9 hours offline to less than 1 minute inline, enabling the onroad emission estimates
simultaneously coupled with the meteorology forecasting. Overall, the MetEmis corrected the low-biases of
NO$_x$ and primary PM$_{2.5}$ emissions domain wide, and high-biases of VOC emissions in California. The
MetEmis also successfully captured the temporal variation of onroad vehicle emissions, resulted in an
improved simulated NO$_2$, O$_3$ and PM$_{2.5}$ concentrations with more agreement with observations compared to
the ones using static temporal profiles. Particularly, the simulated NO$_2$ concentration exhibits noticeable
improvement with increased R$^2$ and decreased RMSEs in most cities. The simulated O$_3$ and PM$_{2.5}$
concentrations were also improved, particularly in summer.
The newly developed CMAQ-MetEmis model demonstrates the importance of dynamic-coupling emissions
and meteorological forecasting. While this study only focused on the onroad emissions, other meteorology-
induced sectors such as residential combustions and agricultural livestock are planned to be included in the
MetEmis development to well represent the meteorological influence on all meteorologically-induced
anthropogenic emissions.




**Digital Object Identifier (DOI) for the CMAQ-MetEmis Coupler:**
https://doi.org/10.5281/zenodo.7150000

**Code Availability:**
The source codes of the SMOKE and the CMAQ models for MetEmis coupler can be downloaded from the DOI website (https://doi.org/10.5281/zenodo.7150000)

**Data availability:**
All the datasets, excel and python scripts used in this manuscript for the data analysis are uploaded through the DOI website (https://doi.org/10.5281/zenodo.7150000)

**Author contribution**
Dr. B.H. Baek is the lead researcher in this study, and Drs. Baek and Coats developed the source codes of CMAQ-MetEmis. Drs. Ma, Wang, Xing and Tong prepared the modeling inputs and analyzed the modeling results. Drs. Woo and Kim participated in the design of the weather-aware emission modeling system.

**Competing interests**
The Authors declare that they have no conflict of interest.

**Acknowledgments**
This research was funded by the National Oceanic and Atmospheric Administration (NOAA)'s Office of Weather and Air Quality (OWAQ) to improve the National Air Quality Forecasting Capability (NAQFC) (Award: NOAA-OAR-OWAQ-2019-2005820) and National Strategic Project-Fine Particle of the National
Research Foundation (NRF) of Korea funded by the Ministry of Science and ICT (MSIT), the Ministry of Environment (ME), the Ministry of Health and Welfare (MOHW) (NRF-2017M3D8A1092022), and by the Korea Environmental Industry & Technology Institute (KEITI) through the Public Technology Program based on Environmental Policy Program, funded by Korea Ministry of Environment (MOE) (2019000160007).

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





## Tables

Table 1. CMAQ modeling domain and configurations.

|  | *Base* | *MetEmis* |
|---|---|---|
| Horizontal Resolution | 12km x 12km | |
| Meteorology | WRFv4.0 with GFS acting as ICs/BCs, RRTMG short/long wave scheme, Noah-MP land-surface scheme, YSU boundary layer scheme | |
| Boundary Condition | GEOS monthly product | |
| Initial Condition | CMAQ restart file | |
| Chemistry | CMAQv5.3.2 CB6r3 AE7 | |
| Emissions | 2017 NEI: Onroad monthly emissions | 2017 NEI: Onroad Meteorology-induced emissions |


Table 2. The required computational memory and time in the SMOKE modeling system.

| Sector | Individual File Size | Total File Size (668 counties) | CPU Memory Usage (GB) | CPU Computing Time* |
|---|---|---|---|---|
| RPD | 50-160 MB | 62.8 GB | 10~20 | ~ 90 mins/day |
| RPV | 26-89 MB | 34.5 GB | 5~10 | ~ 18 mins/day |
| RPH | 7-94 KB | 43.6 MB | 1~2 | ~ 1 mins/day |

* The specification of CPU is Intel Xeon Gold 6240R @ 2.4GHz


Table 3. Statistical metrics between observed and simulated $O_3$, $NO_2$ and $PM_{2.5}$ in January and July, 2019 over contiguous United States

| | January 2019 | | | | | | July 2019 | | | | | |
|---|---|---|---|---|---|---|---|---|---|---|---|---|
| | $O_3$ | | $NO_2$ | | $PM_{2.5}$ | | $O_3$ | | $NO_2$ | | $PM_{2.5}$ | |
| | Base | MetEmis | Base | MetEmis | Base | MetEmis | Base | MetEmis | Base | MetEmis | Base | MetEmis |
| CORR | 0.51 | 0.51 | 0.64 | 0.64 | 0.46 | 0.46 | 0.64 | 0.64 | 0.51 | 0.51 | 0.38 | 0.38 |
| RMSE | 7.03 | 7.00 | 8.33 | 8.27 | 5.72 | 5.76 | 9.56 | 9.51 | 5.69 | 5.67 | 5.03 | 5.04 |
| NMB | -0.01 | -0.01 | -0.32 | -0.30 | 0.10 | 0.11 | -0.01 | -0.01 | -0.15 | -0.15 | -0.05 | -0.05 |
| NME | 17% | 17% | 52% | 52% | 46% | 47% | 17% | 17% | 62% | 62% | 40% | 40% |







Table 4. The largest differences of ozone episodes in July 2019 over the U.S.

| Episodes | Date @ Time | Base (ppb) | MetEmis (ppb) | Location |
|---|---|---|---|---|
| Largest Increase | Jul 5, 2019 @ 1PM | 78.3 | 85.9 (+7.1) | Chicago, IL |
| Largest Decrease | Jul 24, 2019 @ 11AM | 112.9 | 31.0 (-81.9) | San Jose, CA |


Table 5. Summary of precursor (NOx and VOC) concentrations in the morning before the largest ozone increase episode at 14LST on July 5th, 2019 over Chicago, IL.

| Jul 5th, 2019 | NOx (ppb) | | | | VOC (ppbC) | | | |
|---|---|---|---|---|---|---|---|---|
| | Time | Base | MetEmis | Diff (M-B) | Time | Base | MetEmis | Diff (M-B) |
| Mean | 5-11AM | 8.4 | 8.6 | 0.2 | 5-11AM | 62 | 66 | 4.0 |
| Max | 6-7AM | 18.9 | 20.7 | 1.8 | 6-7AM | 101 | 121 | 20.0 |
| Min | 6-7AM | 8.5 | 8.2 | -0.3 | 10-11AM | 74 | 73 | -1.0 |


Table 6. Statistics of largest ozone decrease episode (July 24th, 2019) over San Jose, CA.

| Jul 24th, 2019 | NOx (ppb) | | | | VOC (ppbC) | | | 640 |
|---|---|---|---|---|---|---|---|---|
| | Time | Base | MetEmis | Diff (M-B) | Time | Base | MetEmis | Diff (M-B) |
| Mean | 3-9AM | 5.8 | 6.8 | 1.0 | 3-9AM | 184 | 35 | 148 |
| Max | 10-11AM | 9.0 | 22.0 | 13.0 | 8-9AM | 1263 | 68 | -1195 |
| Min | 11-12pM | 10.8 | 10.6 | -0.2 | 12PM-1AM | 7.8 | 7.3 | -0.5 |

# Figures

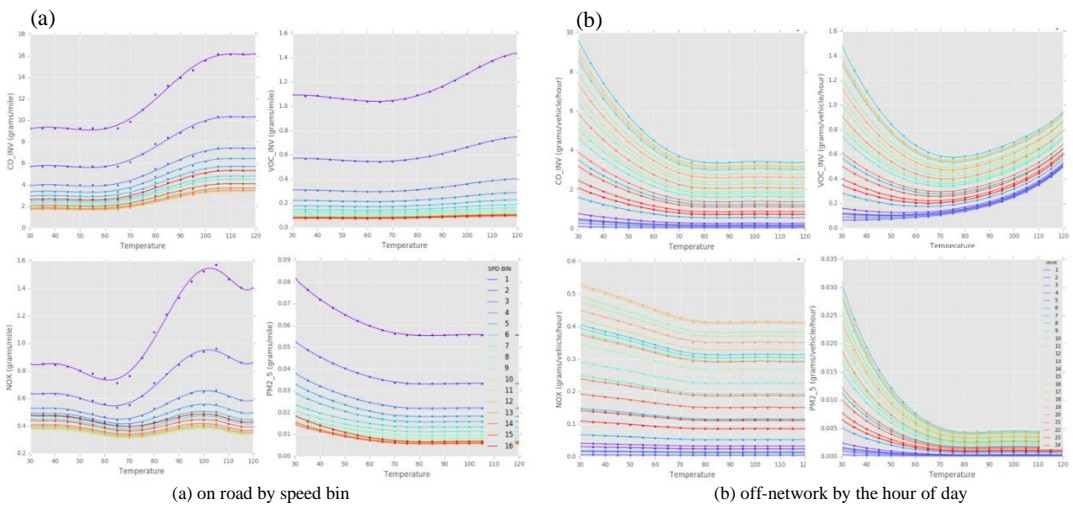

(a) on road by speed bin          (b) off-network by the hour of day

Figure 1. Meteorology-dependency of CO, VOC, NOx, and PM$_{2.5}$ emissions from gasoline-fueled light-duty vehicles by average speed bin (a), and the off-network by the hour of day (b).



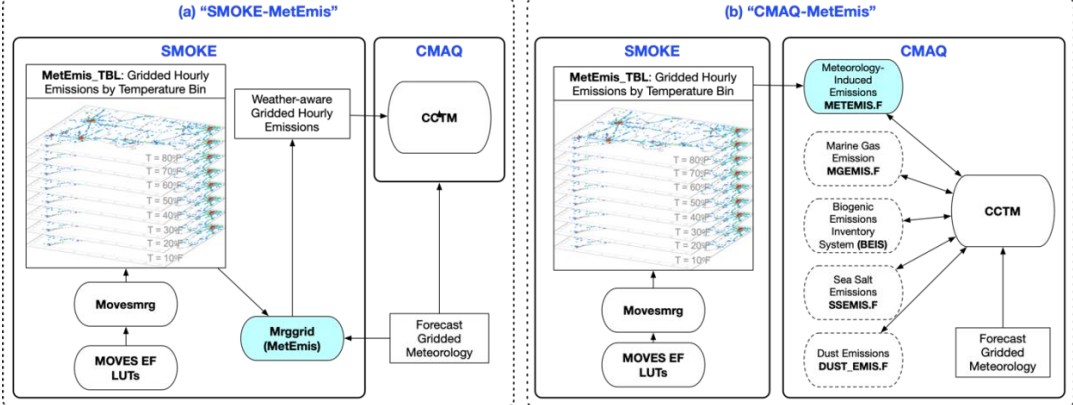

Figure 2. Meteorological-Induced Emissions coupler module "MetEmis" with air quality modeling system: a) "SMOKE-MetEmis", and b) "CMAQ-MetEmis".



Figure 3. Spatial comparison of monthly total emissions of VOC, NO, and PM$_{2.5}$. The colors indicate the MetEmis is larger than Base (red) or smaller (blue) for (a) VOC in January, (b) VOC in July, (c) NO$_X$ in January, (d) NO$_X$ in July,(e) PM$_{2.5}$ in January and (f) PM$_{2.5}$ in July.


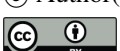

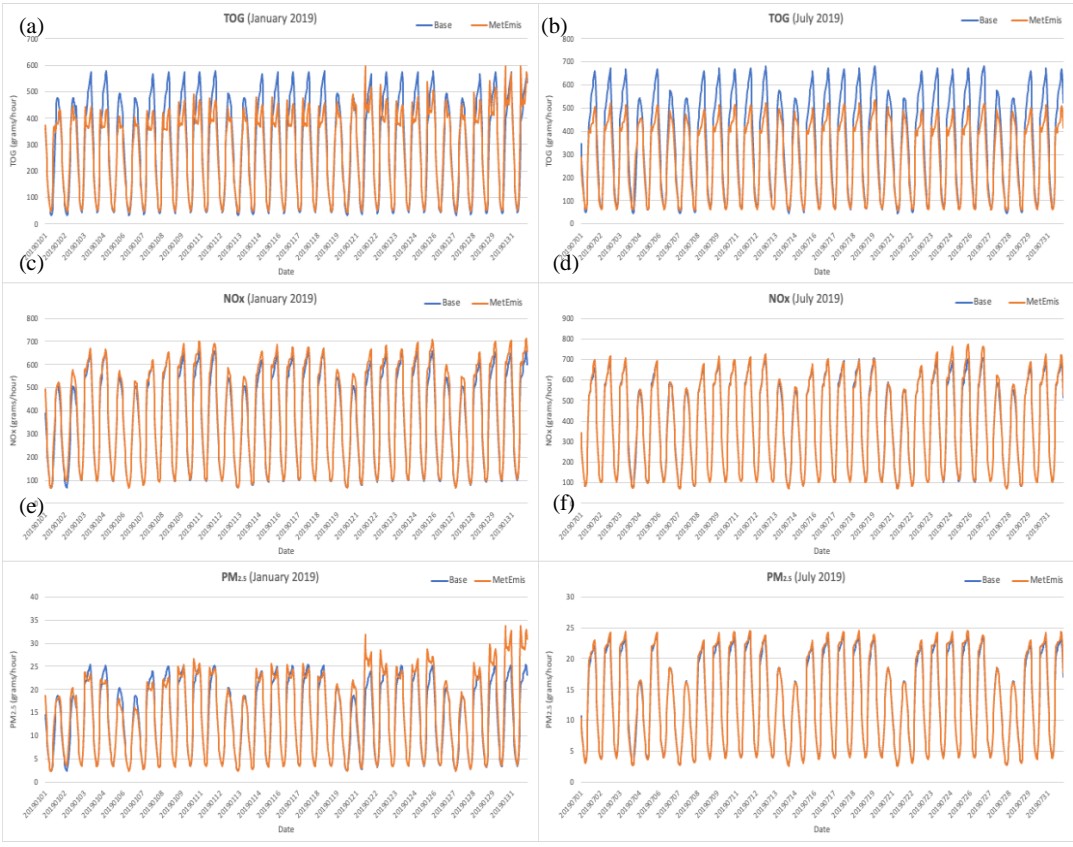

Figure 4. Temporal comparisons of daily domain total emissions of (a) Total Organic Gas (TOG) in January, (b) TOG in July, (c) $NO_X$ in January, (d) NOx in July,(e) $PM_{2.5}$ in January and (f) $PM_{2.5}$ in July from the Base (blue line) and MetEmis scenarios (red line).







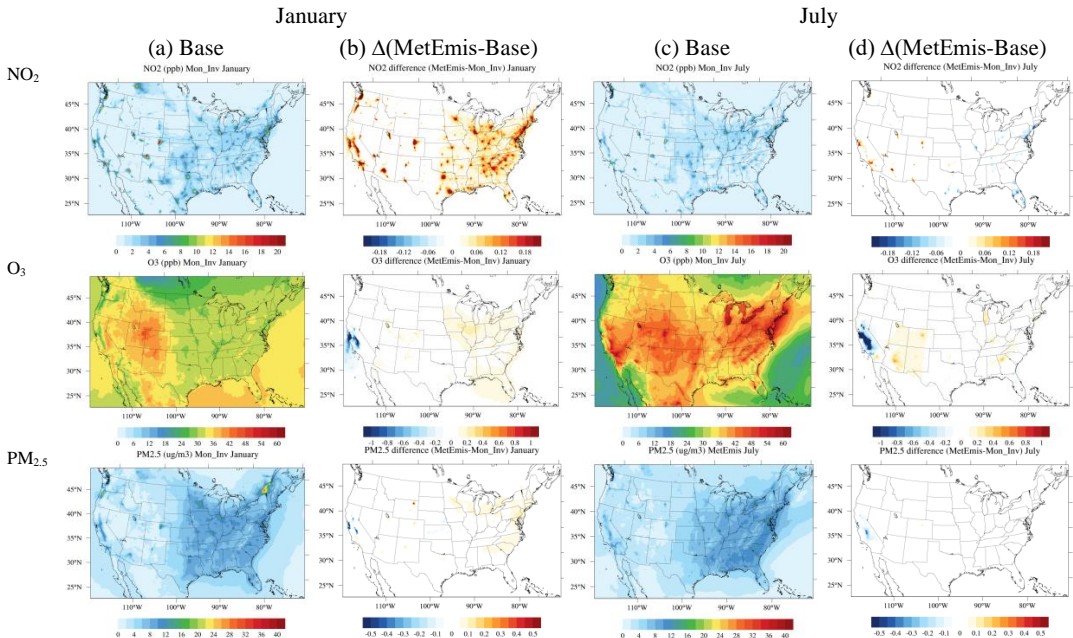

Figure 5. spatial distribution of $NO_2$, $O_3$ and $PM_{2.5}$ concentrations and difference figures: (a) January averaged concentrations from Base scenario, (b) the differences between Base and MetEmis scenarios in January, (c) July averaged concentrations from Base scenario, and (d) the differences between Base and
MetEmis scenarios in July









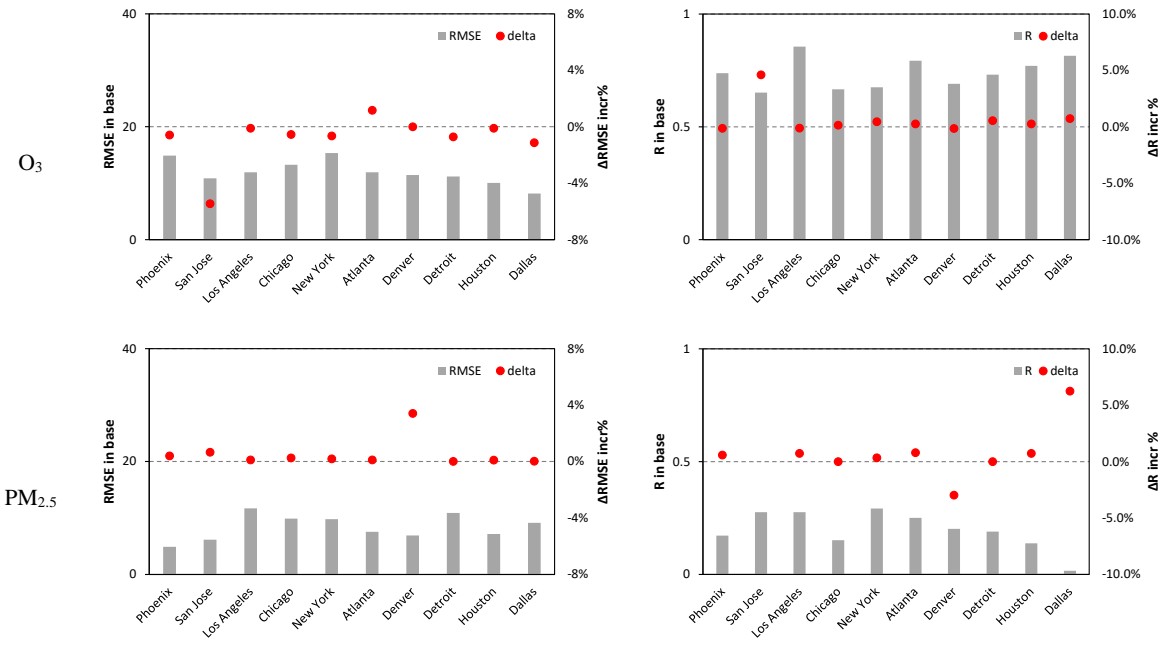

Figure 6. Comparison of model performance in simulating $NO_2$, $O_3$ and $PM_{2.5}$ concentrations between Base and MetEmis scenarios. The columns panels shows the different model evaluation metrics in January (panel a and b) and July (panel c and d). The rows present different species including $NO_2$, $O_3$, and $PM_{2.5}$. RMSE is Root-mean-square deviation, R is correlation coefficient. delta is (MetEmis - Base)/Base; when $\Delta R > 0$ and $\Delta RMSE < 0$, indicate the improvement in MetEmis.

* $NO_2$ in January in Denver is -0.002, increased to 0.008 with MetEmis; Observed $O_3$ data is missing in Chicago and Atlanta in January.



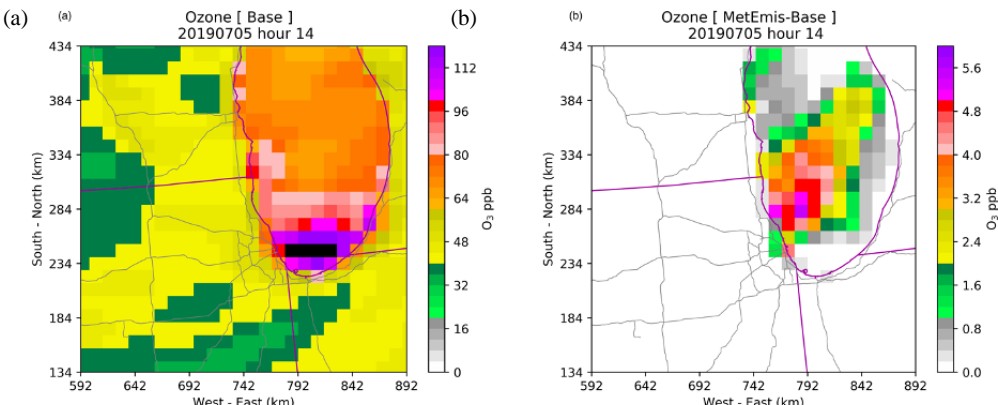


Figure 7. Base hourly ozone (ppb) (a) and the hourly ozone difference (MetEmis-Base) (b) at 14LST on July 5th, 2019. Black color indicates the concentration above the color scale maximum (120 ppb)



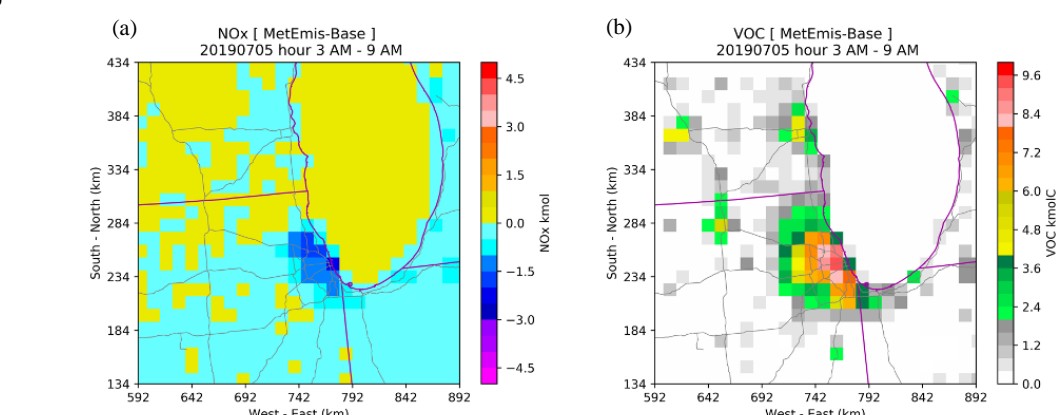

Figure 8. Spatial differences of NOx (a) and VOC (b) emissions in early morning (3AM-9AM) on Jul 5th, 2019.





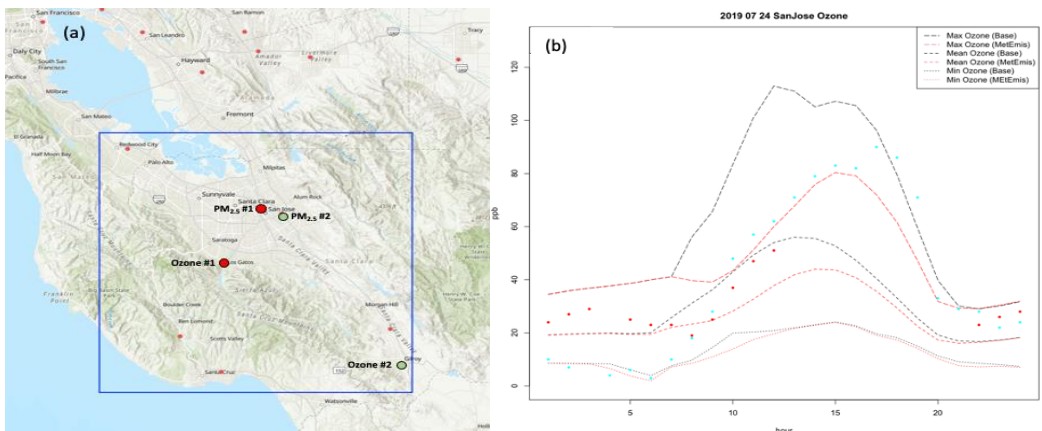

Figure 9. (a) U.S. EPA's Air Quality System (AQS) ozone and PM2.5 monitoring locations, and (b) diurnal variation of ozone (maximum, mean and minimum) on July 24, 2022 over San Jose, CA. The base map layer of this figure was made by Esri (Esri, 2013)



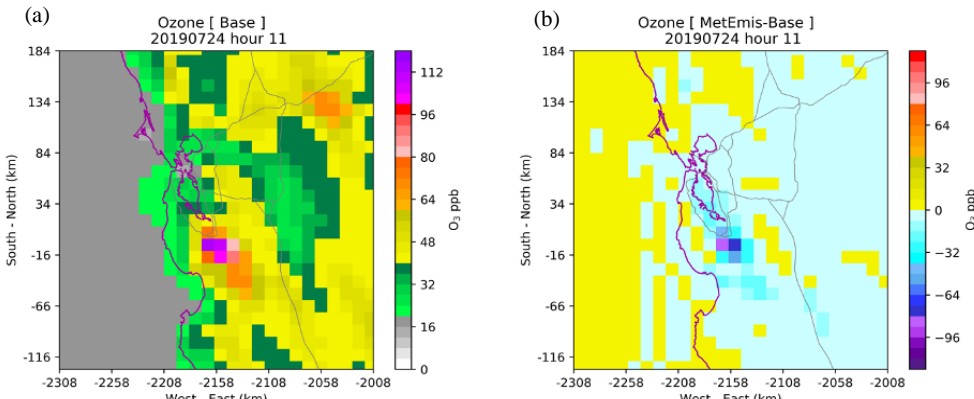

Figure 10. Base hourly ozone concentration (ppb) (a) and the hourly ozone difference (MetEmis-Base) (b) at 11LST on July 24th, 2019.






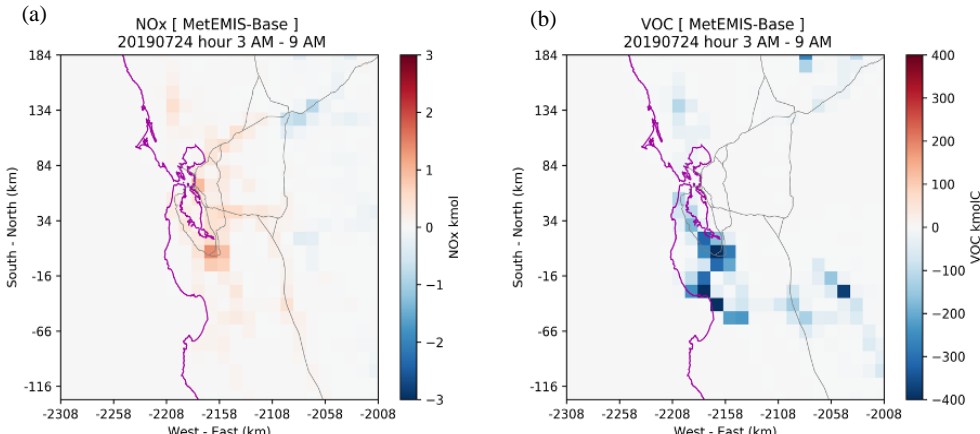

Figure 11. Spatial differences of NOx (a) and VOC (b) emissions from 3LST to 9LST on July 24, 2019 over San Jose, CA.



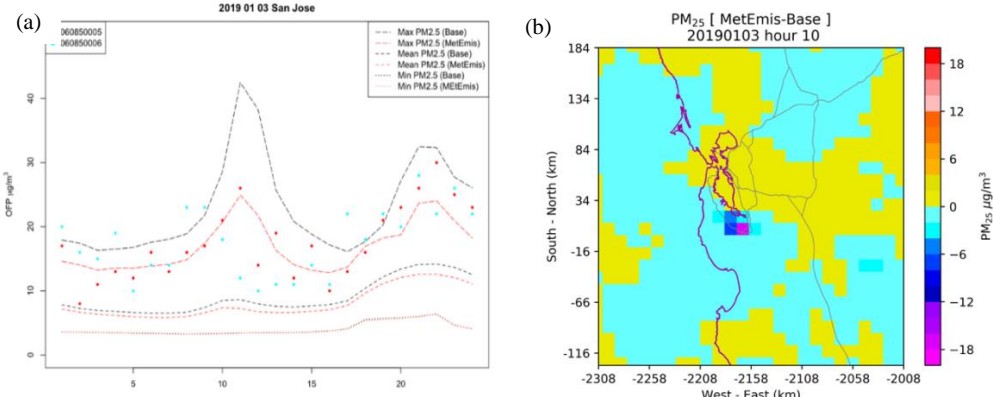

Figure 12. (a)Diurnal variation of PM$_{2.5}$ (maximum, mean and minimum) concentrations over San Jose targeted region, and (b) the spatial difference of PM2.5 at 10LST on January 3, 2019.






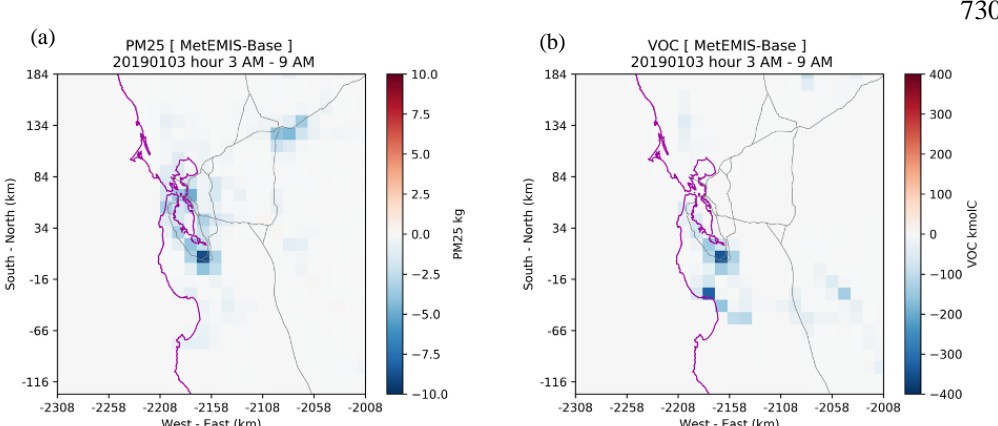

Figure 13. Spatial difference of PM$_{2.5}$ (a) and VOC (b) emissions over San Jose region from
3LST to 9LST on January 3, 2019.