# Peer review of "Dynamic Meteorology-Induced Emissions Coupler (MetEmis) development in the Community Multiscale Air Quality (CMAQ): CMAQ-MetEmis"

_Geoscientific Model Development, 2022_

## Author Comment (AC1)

**Responses to the Reviewer's Comments/Suggestions**

**May 3, 2023**

We thank our reviewers for their comments regarding the importance and timeliness of our study. The original reviewer's comments/suggestions are in **black**, and the author's responses are in **blue.**

**Reviewer #1 comments and suggestions**

This manuscript describes the development of a new module which realizes online consideration of the influences of meteorological conditions on emissions used in air quality simulations. An application of this module on vehicle emissions and air quality simulations is also described.

I can understand the importance of considering the influences of meteorological conditions on emissions. However, the descriptions are very confusing for me. I have difficulties understanding what was done in this study and what is the scientific significance of the module developed in this study.

MOVES is a model to estimate vehicle emissions as described in the first paragraph in Section 2.3. Its computational requirements are prohibitive in real-time air quality forecasting applications. Therefore, the SMOKE-MOVES tool was developed to overcome the issues. It runs SMOKE to estimate air quality model-ready emissions using the MOVES EF LUTs with hourly meteorological inputs as described in the second paragraph in Section 2.3. However, MOVES EF LUT files require significant computational resources, such as memory and storage spaces as described in the third paragraph.

According to Figure 2, MOVES EF LUTs are the starting points in SMOKE-MetEmis and CMAQ-MetEmis. Therefore, I thought that SMOKE-MOVES is inevitable even if MOVES EF LUT files require significant computational resources

However, the fourth paragraph of Section 3.1 says that the SMOKE-MetEmis can generate a single MetEmis_TBL emissions input file that can represent the 334 MOVES LUTs files and its size is significantly smaller than the size of the 334 MOVES LUTs files.

Does it mean the new module can generate a single MetEmis_TBL emissions input file which can be used instead of 334 MOVES LUTs files? But I cannot find how the new module can generate a single MetEmis_TBL emissions input file without using SMOKE-MOVES and MOVES LUTs files.

**Response:** Thanks for your comments. Your points on the confusion between SMOKE-MetEmis and CMAQ-MetEmis and restructuring are well taken. So, we made some significant changes to address your points.

We can clearly understand that it can be confusing to understand the differences between the current SMOKE-MOVES and these two new MetEmis coupler modules (SMOKE-MetEmis and CMAQ-MetEmis) due to the lack of clarification in the original manuscript. To clarify the differences between these approaches, we have restructured Section 2 and 3 by adding a new section, and paragraphs to describe the differences and their pros/cons in the revised manuscript. As stated in Section 2.2, the SMOKE-MOVES tool was developed to dynamically estimate onroad mobile emissions based on the 334 MOVES EF LUT files and simulated meteorology. This SMOKE-MOVES tool was developed back in 2010 and designed for offline dynamic onroad mobile estimations due to the significant computational resource required for MOVES model itself, which takes up to 1.5 day for MOVES simulation run in the "Emission Rates Mode" (described in Section 2.2). To overcome this computational bottleneck issue in MOVES, the SMOKE-MOVES tool was introduced by the author and the team of EPA inventory group to process these complex emission factors from MOVES through the SMOKE modeling system. However, due to the size of emission factors output files from the MOVES, the SMOKE-MOVES tool still requires a significant amount of computational resources as described in Section 2.4. Thus, we have developed the inline coupler called "MetEmis" for meteorology-induced emission sources like onroad mobile vehicles to significantly expedite the processing time and then it can be coupled with the CMAQ model as one of the "inline" modules (e.g., lightening NOx, biogenic emissions, sea salt, plume rise calculation, and more). The "MetEmis" coupler has been implemented in both SMOKE emissions model (SMOKE-MetEmis) and CMAQ air quality model (CMAQ-MetEmis) to enable dynamic inline coupling for the NAQFC forecasting application.

To bring the clarity of MetEmis coupler approaches (SMOKE-MetEmis and CMAQ-MetEmis), we have clearly stated that these new MetEmis couplers produce the same gridded hourly onroad mobile emissions as the ones from the original SMOKE-Integration tool to meet the requirement of replication in the new Section 2.3. We did not bring any additional uncertainties through our MetEmis approaches but exactly mimicked the original approach with a significant computational speed and enabled us to dynamically estimate the onroad mobile emissions within the CMAQ modeling system to reflect the indirect/direct aerosol feedback to local/regional simulated meteorology.

In new Section 2.4, we discussed the main advantage of the MetEmis approach that it is significantly faster computationally than the current SMOKE-MOVES approach and capable of becoming a part of future fully coupled CTM modeling system without losing any accuracy against the offline SMOKE-MOVES approach.

As stated in the Section 2.2, all MOVES EF LUTs are the input files to the SMOKE-MOVES integration tool currently used in the US EPA emission modeling platform to generate the CMAQ-ready gridded hourly emissions. Because of the number and size of the MOVES EF LUTs, SMOKE-MOVES approach requires a significant computing memory and spaces to process them offline. The new MetEmis coupler option in SMOKE (SMOKE-MetEmis) can well

overcome such limitations by using a single MetEmis_TBL emission input file to consider the variation emission factors corresponding to the different meteorological conditions, similar to the original 334 MOVES LUTs.

The SMOKE-MOVES integration tool in the upcoming SMOKE v5.0 has been updated to optionally generate the MetEmis_TBL output files to support these MetEmis coupler modules (SMOKE-MetEmis and CMAQ-MetEmis).

**Reviewer's Comment:** It is important to consider influences of meteorological conditions on emissions. However, if their influences are poorly represented, the model performance could be also poorer when they are considered. Therefore, accuracies of influences of meteorological conditions are critical in term of this study. However, as described in Line 171-175, the dependency of mobile emissions on local meteorology can vary by vehicle types, fuel types, road types, processes, vehicle speed for onroad vehicles, hour of day for off-network vehicles, as well as by pollutants. Uncertainties in them could be quite large. Nevertheless, this manuscript just believed the dependence of vehicle emissions on meteorological conditions represented in MOVES. Their uncertainties should be discussed.

**Response:** Thanks for your comment on this issue. As stated in Section 2.3, the MetEmis coupler can produce identical results from the current SMOKE-MOVES integration tool without any computation resources, there are no new uncertainties introduced in the MetEmis coupler development. The native uncertainties of onroad mobile emissions are from the US EPA's MOVES model, not from the MetEmis modules. This manuscript focuses on replicating the same dynamic emissions from the offline SMOKE-MOVES onroad mobile emissions for the CMAQ model as the inline option. So, the native uncertainties from the MOVES model are not discussed in this manuscript. Following the reviewer's suggestion, we have clarified that the native uncertainties from MOVES should also be noted and reduced in future studies.

**Reviewer's Comment:** Figure 4 (b) indicates that TOG emissions estimated in "Base" is mostly higher than "MetEmis". This is not due to an issue of "online" or "offline". The profile used in "Base" may not be just representative. The performance could become higher if a more representative meteorological profile is provided.

**Response:** Thanks for your comment. We improve the distinction between "Base" and "MetEmis" by moving the definition of these two scenarios ("Base" and "MetEmis") into the beginning of the Section 3 "Results" and then added a paragraph to clearly state the focus on modeling applications and evaluations between the static offline "Base" and the dynamic inline "MetEmis" scenarios.

As you stated correctly, the TOG differences from two different scenarios are not caused by "inline" and "offline" but by the lack of temporal representation of "Base" emissions from the MOVES "Inventory Mode" simulation. That is why we believe the "MetEmis" coupler should be implemented to mimic the state-of-the-art SMOKE-MOVE model to well represent the meteorological profiles with limited requirement of computational resources.

**Reviewer's Comment:** Then, improvement in model performance with the new module is quite small as shown in Table 3. Differences between "Base" and "MetEmis" are within uncertainties in concentrations simulated by chemical transport models.

**Response:** Because we mostly updated the onroad mobile emissions from surrounding the metropolitan cities, we do not see much of MetEmis impacts in the rural area where the onroad mobile emissions impacts on local air quality are limited. We divided Section 3 into the general domain-level evaluation and the city-level evaluation.

While there seem to be no significant local meteorology impacts on local air quality, we demonstrated in the Section 3.2 that most impacts occur at the city-level near the metropolitan area where the onroad mobile emissions play a critical role in local air quality. Especially, Figure 9 demonstrated that the peak of ozone could be improved by using the dynamically-estimated hourly emissions with simulated meteorology.

**Reviewer's Comment:** Large differences can be seen only in the selected episodes. In case of the episode (July 24th, 2019) over San Jose, CA, the maximum VOC concentration is 1263 ppbC as shown in Table 6. This unrealistic high value could be originated in errors in the meteorological profiles used in "Base".

**Response:** As stated in the manuscript, these large differences are caused by the emissions from two different static "Base" and dynamic "MetEmis" scenarios. The main difference between these scenarios is from spatiotemporal emissions due to the MOVES emission factors as well as local meteorology from "MetEmis" scenario compared to the temporally static emission from the "Base" scenario. Due to the substantial requirement of computational resources of SMOKE-MOVES, most systems (like NOAA- NAQFC) have to rely on static temporal profiles like "Base" thus suffering large uncertainties. That is why the "MetEmis" proposed in this study is so important to be implemented in the forecasting system.

**Reviewer #2 comments and suggestions**

This manuscript describes the CMAQ-MetEmis module, which allows to dynamically model the meteorological-induced MOVES onroad mobile emissions inline within the CMAQ air quality modelling system. The CMAQ-MetEmis module addresses the shortcomings (computational time and memory requirements) of the current "offline" approach, which is based on the SMOKE model. The strength of CMAQ-MetEmis is in its ability to take into account the impact of local meteorology on mobile emissions within a reasonable time frame to meet the air quality forecasting time constraints. The manuscript shows how the spatiotemporal enhancements of on-road mobile emissions predicted by "MetEmis" benefit the performance of an air quality modelling system, indicating the importance of dynamically estimate weather-aware mobile emissions. The paper is well written and structured, and its quality is good, which makes it a very good contribution to GMD. I therefore recommend to accept this manuscript for publication once the following comments have been addressed.

1. The meteorological-dependent expressions considered in this study are key to understand the emission and air quality results that are later presented. From my point of view these expressions should be included and briefly described in the manuscript, including also a discussion on the potential uncertainty associated to them.

   **Response:** Thanks for your comment on this issue. As stated in the Section 2.3, the MetEmis coupler can produce identical results from the current SMOKE-MOVES integration tool without any computation resources, there are no new uncertainties introduced in the MetEmis development. The native uncertainties of onroad mobile emissions are from the US EPA's MOVES model, not from the MetEmis modules. As the reviewer suggested, we have clarified that the native uncertainties from MOVES should also be noted and reduced in future studies.

2. Despite reporting interesting results, the contents of sections 3.3 and 3.4 are sometimes a bit difficult to follow (and not always in line with the proposed titles). I would recommend to restructure this part of the manuscript by: 1) creating a general section entitled e.g., "Effects of Weather-Aware Mobile Emissions on modeling performance", and then including several subsection describing the differences between the base and MetEmis cases 1) on a general level, 2) at the city level and 3) for specific pollution episodes. Despite reporting a general improvement, authors should emphasize a bit more that these tend to only be relevant for NO2, and highlight that improving statistics of secondary pollutants (O3) or pollutants that mainly consist on secondary species (PM2.5) is more challenging due to the important role of other emission sources and processes (e.g., chemical reactions).

   **Response:** Thanks for your suggestions on restricting the result section. We followed your suggestion to subdivide the CTM evaluation sections from overall, cities, and episodic cases. Please check Section 3.2. "Weather-Aware Mobile Emissions Impacts on CTM Simulations"

3. Development of CMAQ-MetEmis Coupler (line 215) ◊ Should not this be header of a subsection?

   **Response:** Thanks for the comment. This header has been removed.

4. Line 250 : "respectively for the winter (January) and winter (July)" (should not be summer?)

   **Response:** Thanks for the comment. The typo has been corrected.

5. Legends in Figure 9b and Figure 12a are difficult to read and incomplete (what are the dots representing?)

   **Response:** Thanks for the comment. The Figure 9b and 12a have been updated with correct legends and bigger fonts to be more clear for readers.